# Data-driven identification of potential Zika virus vectors

**Michelle V Evans[1,2]\*, Tad A Dallas[1,3], Barbara A Han[4], Courtney C Murdock[1,2,5,6,7,8], John M Drake[1,2,8]**

[1]Odum School of Ecology, University of Georgia, Athens, United States; [2]Center for the Ecology of Infectious Diseases, University of Georgia, Athens, United States; [3]Department of Environmental Science and Policy, University of California-Davis, Davis, United States; [4]Cary Institute of Ecosystem Studies, Millbrook, United States; [5]Department of Infectious Disease, University of Georgia, Athens, United States; [6]Center for Tropical Emerging Global Diseases, University of Georgia, Athens, United States; [7]Center for Vaccines and Immunology, University of Georgia, Athens, United States; [8]River Basin Center, University of Georgia, Athens, United States

**Abstract** Zika is an emerging virus whose rapid spread is of great public health concern. Knowledge about transmission remains incomplete, especially concerning potential transmission in geographic areas in which it has not yet been introduced. To identify unknown vectors of Zika, we developed a data-driven model linking vector species and the Zika virus via vector-virus trait combinations that confer a propensity toward associations in an ecological network connecting flaviviruses and their mosquito vectors. Our model predicts that thirty-five species may be able to transmit the virus, seven of which are found in the continental United States, including *Culex quinquefasciatus* and *Cx. pipiens*. We suggest that empirical studies prioritize these species to confirm predictions of vector competence, enabling the correct identification of populations at risk for transmission within the United States.

\*For correspondence: mvevans@uga.edu

## Introduction

In 2014, Zika virus was introduced into Brazil and Haiti, from where it rapidly spread throughout the Americas. By January 2017, over 100,000 cases had been confirmed in 24 different states in Brazil (http://ais.paho.org/phip/viz/ed_zika_cases.asp), with large numbers of reports from many other counties in South and Central America (*Faria et al., 2016*). Originally isolated in Uganda in 1947, the virus remained poorly understood until it began to spread within the South Pacific, including an outbreak affecting 75% of the residents on the island of Yap in 2007 (49 confirmed cases) and over 32,000 cases in the rest of Oceania in 2013–2014, the largest outbreak prior to the Americas (2016-present) (*Cao-Lormeau et al., 2016*; *Duffy et al., 2009*). Guillian-Barré syndrome, a neurological pathology associated with Zika virus infection, was first recognized at this time (*Cao-Lormeau et al., 2016*). Similarly, an increase in newborn microcephaly was found to be correlated with the increase in Zika cases in Brazil in 2015 and 2016 (*Schuler-Faccini et al., 2016*). For this reason, in February 2016, the World Health Organization declared the American Zika virus epidemic to be a Public Health Emergency of International Concern.

Despite its public health importance, the ecology of Zika virus transmission has been poorly understood until recently. It has been presumed that *Aedes aegypti* and *Ae. albopictus* are the primary vectors due to epidemiologic association with Zika virus (*Messina et al., 2016*), viral isolation from and transmission experiments with field populations (especially in *Ae. aegypti* [*Haddow et al.,*

**eLife digest** Mosquitoes carry several diseases that pose an emerging threat to society. Outbreaks of these diseases are often sudden and can spread to previously unaffected areas. For example, the Zika virus was discovered in 1947, but only received international attention when it spread to the Americas in 2014, where it caused over 100,000 cases in Brazil alone. While we now recognize the threat Zika can pose for public health, our knowledge about the ecology of the disease remains poor. Nine species of mosquitoes are known to be able to carry the Zika virus, but it cannot be ruled out that other mosquitoes may also be able to spread the disease.

There are hundreds of species of mosquitoes, and testing all of them is difficult and costly. So far, only a small number of species have been tested to see if they transmit Zika. However, computational tools called decision trees could help by predicting which mosquitoes can transmit a virus based on common traits, such as a mosquito's geographic range, or the symptoms of a virus.

Evans et al. used decision trees to create a model that predicts which species of mosquitoes are potential carriers of Zika virus and should therefore be prioritized for testing. The model took into account all known viruses that belong to the same family as Zika virus and the mosquitoes that carry them. Evans et al. predict that 35 species may be able to carry the Zika virus, seven of which are found in the United States. Two of these mosquito species are known to transmit West Nile Virus and are therefore prime examples of species that should be prioritized for testing. Together, the ranges of the seven American species encompass the whole United States, suggesting Zika virus could affect a much larger area than previously anticipated.

The next step following on from this work will be to carry out experiments to test if the 35 mosquitoes identified by the model are actually able to transmit the Zika virus.

---

*2012*; *Boorman and Porterfield, 1956*; *Haddow et al., 1964*]), and association with related arboviruses (e.g. dengue fever virus, yellow fever virus). Predictions of the potential geographic range of Zika virus in the United States, and associated estimates for the size of the vulnerable population, are therefore primarily based on the distributions of *Ae. aegypti* and *Ae. albopictus*, which jointly extend across the Southwest, Gulf coast, and mid-Atlantic regions of the United States (*Centers for Disease Control and Prevention, 2016*). We reasoned, however, that if other, presently unidentified Zika-competent mosquitoes exist in the Americas, then these projections may be too restricted and therefore optimistically biased. Additionally, recent experimental studies show that the ability of *Ae. aegypti* and *Ae. albopictus* to transmit the virus varies significantly across mosquito populations and geographic regions (*Chouin-Carneiro et al., 2016*), with some populations exhibiting low dissemination rates even though the initial viral titer after inoculation may be high (*Diagne et al., 2015*). This suggests that in some locations other species may be involved in transmission. The outbreak on Yap, for example, was driven by a different species, *Ae. hensilli* (*Ledermann et al., 2014*). Closely related viruses of the *Flaviviridae* family are vectored by over nine mosquito species, on average (see Supplementary Data). Thus, because Zika virus may be associated with multiple mosquito species, we considered it necessary to develop a more comprehensive list of potential Zika vectors.

The gold standard for identifying competent disease vectors requires isolating virus from field-collected mosquitoes, followed by experimental inoculation and laboratory investigation of viral dissemination throughout the body and to the salivary glands (*Barnett, 1960*; *Hardy et al., 1983*), and, when possible, successful transmission back to the vertebrate host (e.g. *Komar et al., 2003*). Unfortunately, these methods are costly, often underestimate the risk of transmission (*Bustamante and Lord, 2010*), and the amount of time required for analyses can delay decision making during an outbreak (*Day, 2001*). To address the problem of identifying potential vector candidates in an actionable time frame, we therefore pursued a data-driven approach to identifying candidate vectors aided by machine learning algorithms for identifying patterns in high dimensional data. If the propensity of mosquito species to associate with Zika virus is statistically associated with common mosquito traits, it is possible to rank mosquito species by the degree of risk represented by their traits – a comparative approach similar to the analysis of risk factors in epidemiology. For instance, a model could be constructed to estimate the statistical discrepancy between the traits of

known vectors (i.e., *Ae. aegypti*, *Ae. albopictus*, and *Ae. hensilli*) and the traits of all possible vectors. Unfortunately, this simplistic approach would inevitably fail due to the small amount of available data (i.e., sample size of 3). Thus, we developed an indirect approach that leverages the information contained in the associations among many virus-mosquito pairs to inform us about specific associations. Specifically, our method identifies covariates associated with the propensity for mosquito species to vector any flavivirus. From this, we constructed a model of the mosquito-flavivirus network and then extracted from this model the life history profile and species list of mosquitoes predicted to associate with Zika virus, which we recommend be experimentally tested for Zika virus competence.

## Results

In total, we identified 132 vector-virus pairs, consisting of 77 mosquito species and 37 flaviviruses. The majority of these species were *Aedes* (32) or *Culex* (24) species. Our supplementary dataset consisted of an additional 103 mosquito species suspected to transmit flaviviruses, but for which evidence of a full transmission cycle does not exist. This resulted in 180 potential mosquito-Zika pairs on which to predict with our trained model. As expected, closely related viruses, such as the four strains of dengue, shared many of the same vectors and were clustered in our network diagram (*Figure 1*). The distribution of vectors to viruses was uneven, with a few viruses vectored by many mosquito species, and rarer viruses vectored by only one or two species. The virus with the most known competent vectors was West Nile virus (31 mosquito vectors), followed by yellow fever virus (24 mosquito vectors). In general, encephalitic viruses such as West Nile virus were found to be more commonly vectored by *Culex* mosquitoes and hemorrhagic viruses were found to be more commonly vectored by *Aedes* mosquitoes (see *Gould and Solomon (2008)* for further distinctions within *Flaviviridae*) (*Figure 1*).

Our ensemble of BRT models trained on common vector and virus traits predicted mosquito vector-virus pairs in the test dataset with high accuracy (AUC = 0.92 ± 0.02; *sensitivity* = 0.858 ± 0.04; *specificity* = 0.872 ± 0.04). Due to non-monotonicity and existence of interactions among predictor variables within our model, one cannot make general statements about the directionality of effect. Thus, we focus on the relative importance of different variables to model performance. The most important variable for accurately predicting the presence of vector-virus pair was the subgenus of the mosquito species, followed by continental range (e.g. continents on which species are present). The number of viruses vectored by a mosquito species and number of mosquito vectors of a virus were the third and fifth most important variables, respectively. Unsurprisingly, this suggests that, when controlling for other variables, mosquitoes and viruses with more known vector-virus pairs (i.e., more viruses vectored and more hosts infected, respectively), are more likely to be part of a predicted pair by the model. Mosquito ecological traits such as larval habitat and salinity tolerance were generally less important than a species' phylogeny or geographic range (*Figure 2*).

When applied to the 180 potential mosquito-Zika pairs, the model predicted thirty-five vectors to be ranked above the threshold (set at the value of the lowest-ranked known vector), for a total of nine known vectors and twenty-six novel, predicted mosquito vectors of Zika (*Table 1*). Of these vectors, there were twenty-four *Aedes* species, nine *Culex* species, one *Psorophora* species, and one *Runchomyia* species. The GBM model's top two ranked vectors for Zika are the most highly-suspected vectors of Zika virus, *Ae. aegypti* and *Ae. albopictus*.

### Model validation

Our supplementary and primary models generally concur and their ranking of potential Zika virus vectors are highly correlated (ρ = 0.508 and ρ = 0.693 on raw and thresholded predictions, respectively). As one might expect, the supplementary model assigned fewer scores of low propensity (*Appendix 1—figure 2*), suggesting that incorporating this additional uncertainty in the training dataset eroded the model's ability to distinguish negative links. The supplementary model's performance on the testing data (AUC = 0.84 ± 0.02), however, indicates that the additional uncertainty did not impede model performance.

When trained on 'leave-one-out' datasets, all three models were able to predict the testing data with high accuracy (AUC = 0.91, AUC = 0.91, AUC = 0.92 for West Nile, dengue, and yellow fever viruses, respectively). Performance varied when models were validated against predictions of 'known

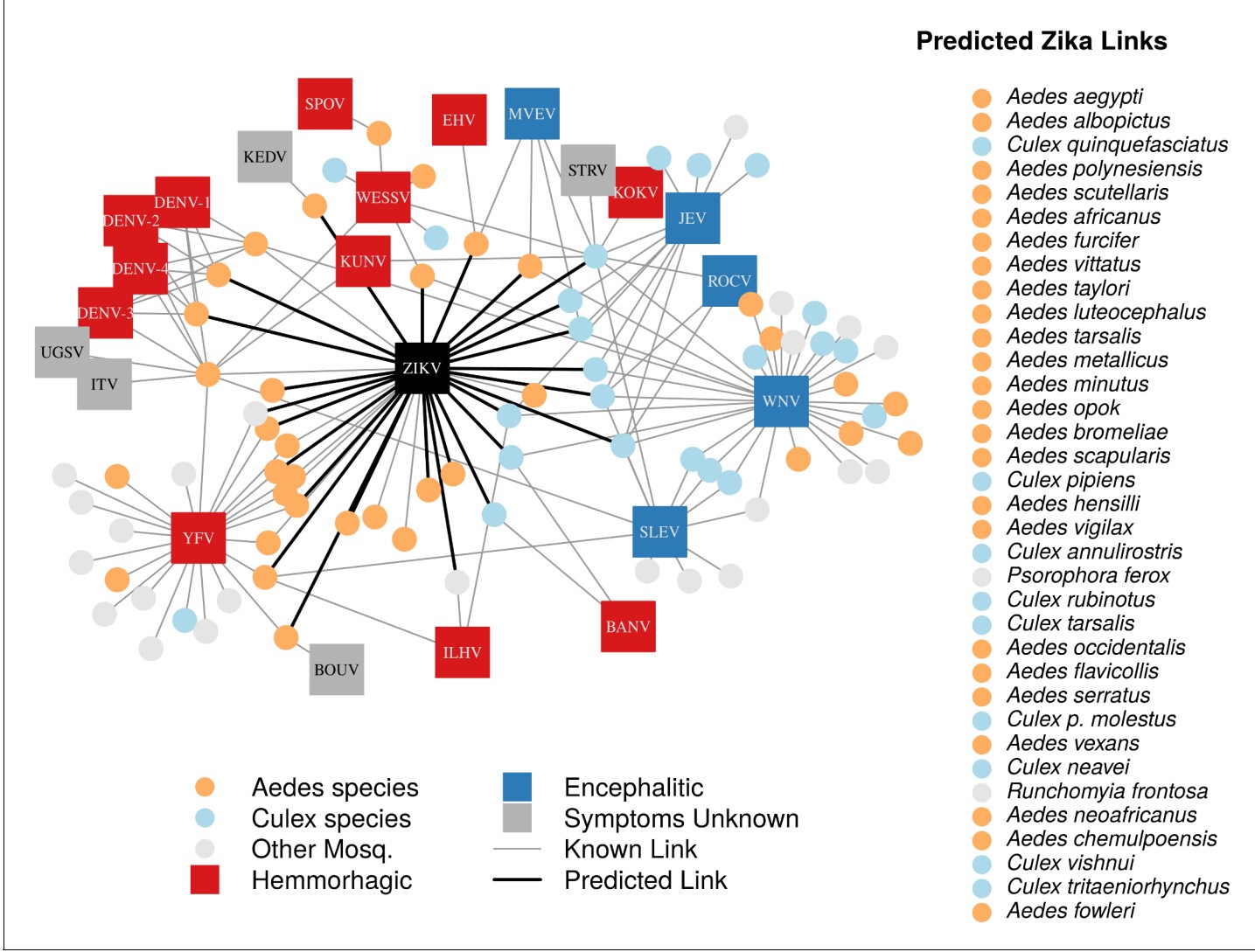

**Figure 1.** A network diagram of mosquito vectors (circles) and their flavivirus pairs (rectangles). The *Culex* mosquitoes (light blue) and primarily encephalitic viruses (blue) are more clustered than the *Aedes* (orange) and hemmorhagic viruses (red). Notably, West Nile Virus is vectored by both *Aedes* and *Culex* species. Predicted vectors of Zika are shown by bolded links in black. The inset shows predicted vectors of Zika and species names, ordered by the model's propensity scores. Included flaviviruses are Banzi virus (BANV), Bouboui virus (BOUV), dengue virus strains 1, 2, 3 and 4 (DENV-1,2,3,4), Edge Hill virus (EHV), Ilheus virus (ILHV), Israel turkey meningoencephalomyelitis virus (ITV), Japanese encephalitis virus (JEV), Kedougou virus (KEDV), Kokobera virus (KOKV), Kunjin virus (KUNV), Murray Valley encephalitis virus (MVEV), Rocio virus (ROCV), St. Louis encephalitis virus (SLEV), Spondwendi virus (SPOV), Stratford virus (STRV), Uganda S virus (UGSV), Wesselsbron virus (WESSV), West Nile Virus (WNV), yellow fever virus (YFV), and Zika virus (ZIKV).

outcomes'. A model trained without West Nile virus predicted highly linked vectors reasonably well (AUC = 0.69), however it assigned low scores to rarer 'known' vectors, such as *Culiseta inornata*, which was only associated with West Nile virus. Similarly, the model trained on the dengue-omitted dataset predicted training data and vectors of dengue itself with high accuracy (AUC = 0.92). While the model trained without yellow fever performed well on the testing data, it performed poorly when predicting vectors of yellow fever virus (AUC = 0.47). Unlike West Nile and dengue viruses, the majority of the known vectors of yellow fever are only associated with yellow fever (i.e. a single vector-virus link), and so were excluded completely from the training data when all yellow fever links were omitted. Additionally, several of the vector species are of the *Haemagogus* genus, which was completely absent from the training data. Given the importance of phylogeny of the vector species in predicting vector-virus links, it follows that a dataset with a novel subgenus would be difficult for

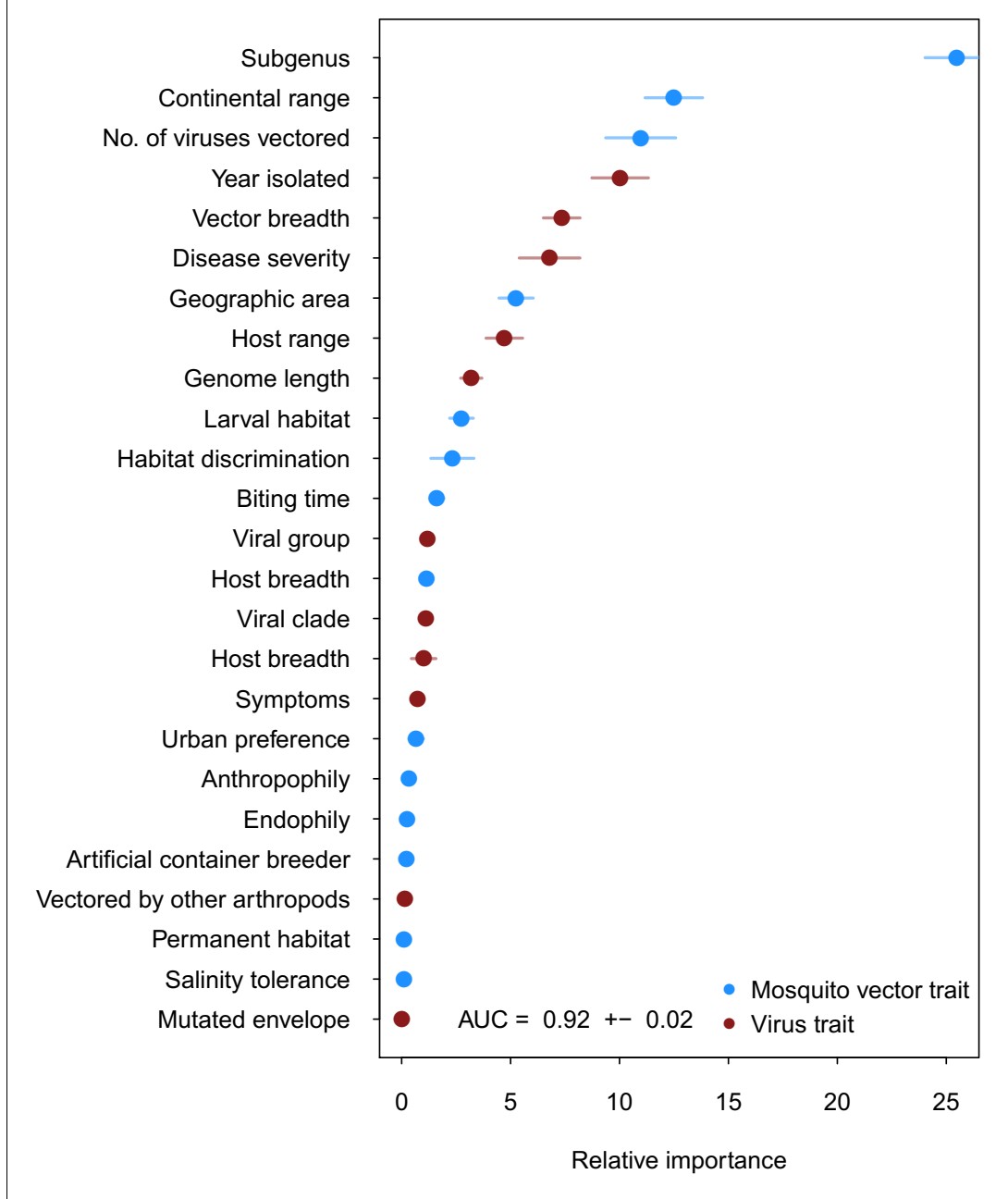

**Figure 2.** Variable importance by permutation, averaged over 25 models. Because some categorical variables were treated as binary by our model (i.e. continental range), the relative importance of each binary variable was summed to result in the overall importance of the categorical variable. Mosquito and virus traits are shown in blue and maroon, respectively. Error bars represent the standard error from 25 models.

the model to predict on, resulting in low model performance. The low performance of this model illustrates that incorporating common traits and additional vector-virus links improves model prediction. When traits were not available in the training dataset, model performance was much lower, suggesting that there exists a statistical association between a vectors' traits and its ability to transmit a virus.

**Table 1.** Predicted vectors of Zika virus, as reported by our model. Mosquito species endemic to the continental United States are bolded. A species is defined as a known vector of Zika virus if a full transmission cycle (see main text) has been observed.

| Species | GBM prediction $\pm SD$ | Known vector? |
|---|---|---|
| *Aedes aegypti* | 0.81 ± 0.12 | Yes |
| *Ae. albopictus* | 0.54 ± 0.14 | Yes |
| *Culex quinquefasciatus* | 0.38 ± 0.14 | No |
| *Ae. polynesiensis* | 0.36 ± 0.13 | No |
| *Ae. scutellaris* | 0.33 ± 0.13 | No |
| *Ae. africanus* | 0.32 ± 0.11 | No |
| *Ae. furcifer* | 0.31 ± 0.16 | Yes |
| *Ae. vittatus* | 0.30 ± 0.20 | Yes |
| *Ae. taylori* | 0.30 ± 0.16 | Yes |
| *Ae. luteocephalus* | 0.25 ± 0.12 | Yes |
| *Ae. tarsalis* | 0.18 ± 0.11 | Yes |
| *Ae. metallicus* | 0.16 ± 0.08 | No |
| *Ae. minutus* | 0.16 ±0.09 | No |
| *Ae. opok* | 0.14 ± 0.06 | No |
| *Ae. bromeliae* | 0.11 ± 0.06 | No |
| *Ae. scapularis* | 0.10 ± 0.04 | No |
| *Cx. pipiens* | 0.10 ± 0.04 | No |
| *Ae. hensilli* | 0.10 ± 0.06 | Yes |
| *Ae. vigilax* | 0.10 ± 0.05 | No |
| *Cx. annulirostrix* | 0.08 ± 0.03 | No |
| *Psorophora ferox* | 0.08 ± 0.05 | No |
| *Cx. rubinotus* | 0.08 ± 0.07 | No |
| *Cx. tarsalis* | 0.08 ± 0.03 | No |
| *Ae. occidentalis* | 0.08 ± 0.05 | No |
| *Ae. flavicolis* | 0.07 ± 0.04 | No |
| *Ae. serratus* | 0.07 ± 0.04 | No |
| *Cx. p. molestus* | 0.07 ± 0.04 | No |
| *Ae. vexans* | 0.06 ± 0.04 | No |
| *Cx. neavei* | 0.06 ± 0.02 | No |
| *Runchomyia frontosa* | 0.06 ± 0.04 | No |
| *Ae. neoafricanus* | 0.06 ± 0.03 | No |
| *Ae. chemulpoensis* | 0.06 ± 0.03 | No |
| *Cx. vishnui* | 0.05 ± 0.01 | No |
| *Cx. tritaeniorhynchus* | 0.05 ± 0.01 | No |
| *Ae. fowleri* | 0.04 ± 0.03 | Yes |

## Discussion

Zika virus is unprecedented among emerging arboviruses in its combination of severe public health hazard, rapid spread, and poor scientific understanding. Particularly crucial to public health preparedness is knowledge about the geographic extent of potentially at risk populations and local environmental conditions for transmission, which are determined by the presence of competent vectors. Until now, identifying additional competent vector species has been a low priority because Zika

virus has historically been geographically restricted to a narrow region of equatorial Africa and Asia (*Petersen et al., 2016*), and the mild symptoms of infection made its range expansion since the 1950's relatively unremarkable. However, with its relatively recent and rapid expansion into the Americas and its association with severe neurological disorders, the prediction of potential disease vectors in non-endemic areas has become a matter of critical public health importance. We identify these potential vector species by developing a data-driven model that identifies candidate vector species of Zika virus by leveraging data on traits of mosquito vectors and their flaviviruses. We suggest that empirical work should prioritize these species in their evaluation of vector competence of mosquitoes for Zika virus.

Our model predicts that fewer than one third of the potential mosquito vectors of Zika virus have been identified, with over twenty-five additional mosquito species worldwide that may have the capacity to contribute to transmission. The continuing focus in the published literature on two species known to transmit Zika virus (*Ae. aegypti* and *Ae. albopictus*) ignores the potential role of other vectors, potentially misrepresenting the spatial extent of risk. In particular, four species predicted by our model to be competent vectors – *Ae. vexans*, *Culex quinquefasciatus*, *Cx. pipiens*, and *Cx. tarsalis* – are found throughout the continental United States. Further, the three *Culex* species are primary vectors of West Nile virus (*Farajollahi et al., 2011*). *Cx. quinquefasciatus* and *Cx. pipiens* were ranked 3rd and 17th by our model, respectively, and together these species were the highest-ranking species endemic to the United States after the known vectors (*Ae. aegypti* and *Ae. albopictus*). *Cx. quinquefasciatus* has previously been implicated as an important vector of encephalitic flaviviruses, specifically West Nile virus and St. Louis encephalitis (*Turell et al., 2005*; *Hayes et al., 2005*), and a hybridization of the species with *Cx. pipiens* readily bites humans (*Fonseca et al., 2004*). The empirical data available on the vector competence of *Cx. pipiens* and *Cx. quinquefasciatus* is currently mixed, with some studies finding evidence for virus transmission and others not (*Guo et al., 2016*; *Aliota et al., 2016*; *Fernandes et al., 2016*; *Huang et al., 2016*). These results suggest, in combination with evidence for significant genotype x genotype effects on the vector competence of *Ae. aegypti* and *Ae. albopictus* to transmit Zika (*Chouin-Carneiro et al., 2016*), that the vector competence of *Cx. pipiens* and *Cx. quinquefasciatus* for Zika virus could be highly dependent upon the genetic background of the mosquito-virus pairing, as well as local environmental conditions. Thus, considering their anthropophilic natures and wide geographic ranges, *Cx. quinquefasciatus* and *Cx. pipiens* could potentially play a larger role in the transmission of Zika in the continental United States. Further experimental research into the competence of populations of *Cx. pipiens* to transmit Zika virus across a wider geographic range is therefore highly recommended, and should be prioritized.

The vectors predicted by our model have a combined geographic range much larger than that of the currently suspected vectors of Zika (*Figure 3*), suggesting that, were these species to be confirmed as vectors, a larger population may be at risk of Zika infection than depicted by maps focusing solely on *Ae. aegypti* and *Ae. albopictus*. The range of *Cx. pipiens* includes the Pacific Northwest and the upper mid-West, areas that are not within the known range of *Ae. aegypti* or *Ae. albopictus* (*Darsie and Ward, 2005*). Furthermore, *Ae. vexans*, another predicted vector of Zika virus, is found throughout the continental US and the range of *Cx. tarsalis* extends along the entire West coast (*Darsie and Ward, 2005*). On a finer scale, these species use a more diverse set of habitats, with *Ae. aegypti* and *Cx. quinquefasciatus* mainly breeding in artificial containers, and *Ae. vexans* and *Ae. albopictus* being relatively indiscriminate in their breeding sites, including breeding in natural sites such as tree holes and swamps. Therefore, in addition to the wider geographic region supporting potential vectors, these findings suggest that both rural and urban areas could serve as habitat for potential vectors of Zika. We recommend experimental tests of these species for competency to transmit Zika virus, because a confirmation of these vectors would necessitate expanding public health efforts to these areas not currently considered at risk.

While transmission requires a competent vector, vector competence does not necessarily equal transmission risk or inform vectorial capacity. There are many biological factors that, in conjunction with positive vector competence, determine a vector's role in disease transmission. For example, although *Ae. aegypti* mosquitoes are efficient vectors of West Nile virus, they prefer to feed on humans, which are dead-head hosts for the disease, and therefore have low potential to serve as a vector (*Turell et al., 2005*). *Psorophora ferox*, although predicted by our model as a potential vector of Zika virus, would likely play a limited role in transmission because it rarely feeds on humans

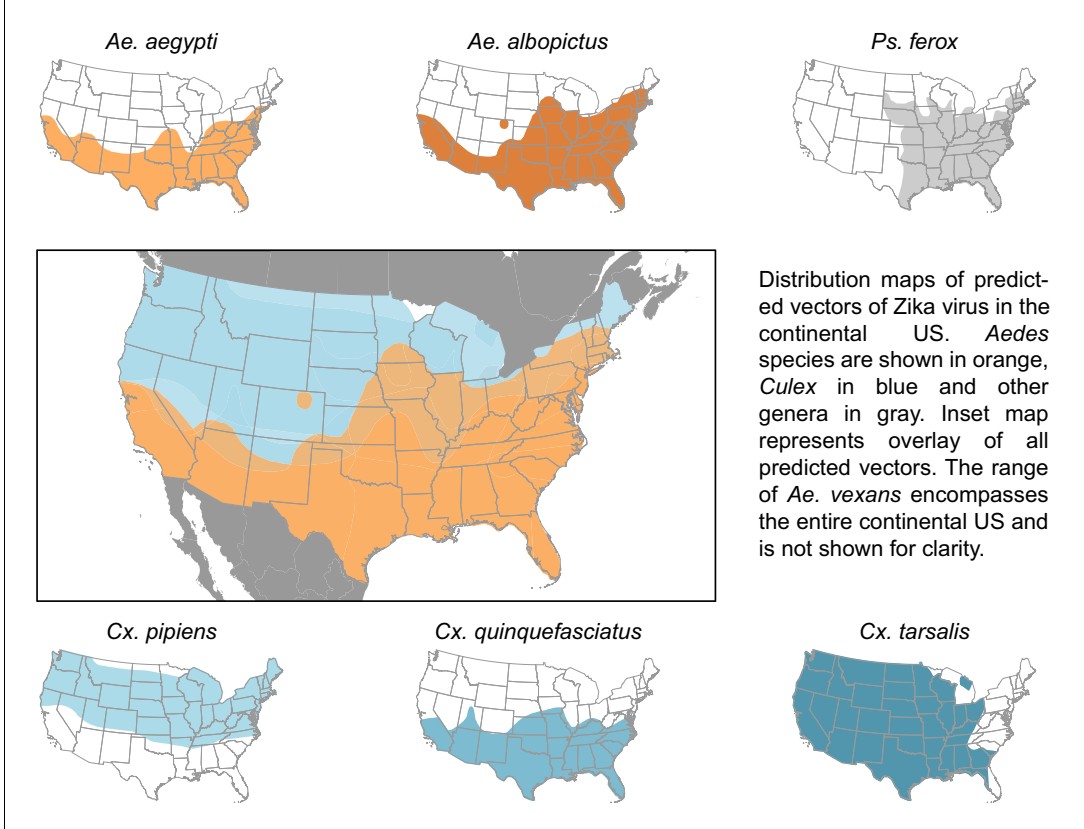

Distribution maps of predicted vectors of Zika virus in the continental US. *Aedes* species are shown in orange, *Culex* in blue and other genera in gray. Inset map represents overlay of all predicted vectors. The range of *Ae. vexans* encompasses the entire continental US and is not shown for clarity.

**Figure 3.** Distribution maps of predicted vectors of Zika virus in the continental US. Maps of *Aedes* species are based on *Centers for disease control and prevention (2016)*. All other species' distributions are georectified maps from *Darsie and Ward (2005)*.

(*Molaei et al., 2008*). Additionally, vector competence is dynamic, and may be mediated by environmental factors that influence viral development and mosquito immunity (*Muturi and Alto, 2011*). Therefore, our list of potential vectors of Zika represents a comprehensive starting point, which should be furthered narrowed by empirical work and consideration of biological details that impact transmission dynamics. Given the severe neurological side-effects of Zika virus infection, beginning with the most conservative method of vector prediction ensures that risk is not underestimated, and allows public health agencies to interpret the possibility of Zika transmission given local conditions.

Our model serves as a starting point to streamlining empirical efforts to identify areas and populations at risk for Zika transmission. While our model enables data-driven predictions about the geographic area at potential risk of Zika transmission, subsequent empirical work investigating Zika vector competence and transmission efficiency is required for model validation, and to inform future analyses of transmission dynamics. For example, in spite of its low transmission efficiency in certain geographic regions (*Chouin-Carneiro et al., 2016*), *Ae. aegypti* is anthropophilic (*Powell and Tabachnick, 2013*), and may therefore pose a greater risk of human-to-human Zika virus transmission than mosquitoes that bite a wider variety of animals. On the other hand, mosquito species that prefer certain hosts in rural environments are known to alter their feeding behaviors to bite alternative hosts (e.g., humans and rodents) in urban settings, due to changes in host community composition (*Chaves et al., 2010*). Environmental factors such as precipitation and temperature directly influence mosquito populations, and determine the density of vectors in a given area (*Thomson et al., 2006*), an important factor in transmission risk. Additionally, socio-economic factors such as housing type and lifestyle can decrease a populations' contact with mosquito vectors, and lower the risk of transmission to humans (*Moreno-Madriñán and Turell, 2017*). Effective risk modeling and forecasting the range expansion of Zika virus in the United States will depend on validating

the vector status of these species, as well as resolving behavioral and biological details that impact transmission dynamics.

Although we developed this model with Zika virus in mind, our findings have implications for other emerging flaviviruses and contribute to the recently developed methodology applying machine learning methods to the prediction of unknown agents of infectious diseases. This technique has been used to predict rodent reservoirs of disease (*Han et al., 2015*) and bat carriers of filoviruses (*Han et al., 2016*) by training models with host-specific data. Our model, however, incorporates additional data by constructing a vector-virus network that is used to inform predictions of vector-virus associations. The combination of common virus traits with vector-specific traits enabled us to predict potential mosquito vectors of specific flaviviruses, and to train the model on additional information distributed throughout the flavivirus-mosquito network.

Uncertainty in our model arises through uncertainty inherent in our datasets. Vector status is not static (e.g. mutation in the chikungunya virus to increase transmission by *Ae. albopictus* [*Weaver and Forrester, 2015*]) and can vary across vector populations (*Bennett et al., 2002*). When incorporating uncertainty in vector status through our supplementary model, our predictions generally agreed with that of our original model. However, the increased uncertainty did reduce the models' ability to distinguish negative links, resulting in higher uncertainty in propensity scores (as measured by standard deviation) and a larger number of predicted vectors. Additionally, the model performs poorly when predicting on vector-virus links with trait levels not included in the training data set, as was the case when omitting yellow fever virus. Another source of uncertainty is regarding vector and virus traits. In addition to intraspecific variation in biological traits, many vectors are understudied, and common traits such as biting activity are unknown to the level of species. Additional study into the behavior and biology of less common vector species would increase the accuracy of prediction techniques such as this, and allow for a better of understanding of species' potential role as vectors.

Interestingly, our constructed flavivirus-mosquito network generally concurs with the proposed dichotomy of *Aedes* species vectoring hemorrhagic or febrile arboviruses and *Culex* species vectoring neurological or encephalitic viruses (*Grard et al., 2010*) (*Figure 1*). However, there are several exceptions to this trend, notably West Nile virus, which is vectored by several *Aedes* species. Additionally, our model predicts several *Culex* species to be possible vectors of Zika virus. While this may initially seem contrary to the common phylogenetic pairing of vectors and viruses noted above, Zika's symptoms, like West Nile virus, are both febrile and neurological. Thus, its symptoms do not follow the conventional hemorrhagic/encephalitic division. The ability of Zika virus to be vectored by a diversity of mosquito vectors could have important public health consequences, as it may expand both the geographic range and seasonal transmission risk of Zika virus, and warrants further empirical investigation.

Considering our predictions of potential vector species and their combined ranges, species on the candidate vector list need to be validated to inform the response to Zika virus. Vector control efforts that target *Aedes* species exclusively may ultimately be unsuccessful in controlling transmission of Zika because they do not control other, unknown vectors. For example, the release of genetically modified *Ae. aegypti* to control vector density through sterile insect technique is species-specific and would not control alternative vectors (*Alphey et al., 2010*). Additionally, species' habitat preferences differ, and control efforts based singularly on reducing *Aedes* larval habitat will not be as successful at controlling *Cx. quinquefasciatus* populations (*Rey et al., 2006*). Predicted vectors of Zika virus must be empirically tested and, if confirmed, vector control efforts would need to respond by widening their focus to control the abundance of all predicted vectors of Zika virus. Similarly, if control efforts are to include all areas at potential risk of disease transmission, public health efforts would need to expand to address regions such as the northern Midwest that fall within the range of the additional vector species predicted by our model. An understanding of the capacity of mosquito species to vector Zika virus is necessary to prepare for the potential establishment of Zika virus in the United States, and we recommend that experimental work start with this list of candidate vector species.

## Materials and methods

### Data collection and feature construction

Our dataset comprised a matrix of vector-virus pairs relating all known flaviviruses and their mosquito vectors. To construct this matrix, we first compiled a list of mosquito-borne flaviviruses to include in our study (*Van Regenmortel et al., 2000*; *Kuno et al., 1998*; *Cook and Holmes, 2006*). Viruses that only infect mosquitoes and are not known to infect humans were not included. Using this list, we constructed a mosquito-virus pair matrix based on the Global Infectious Diseases and Epidemiology Network database (*GIDEON, 2016*), the International Catalog of Arboviruses Including Certain Other Viruses of Vertebrates (ArboCat) (*Karabatsos, 1985*), *The Encyclopedia of Medical and Veterinary Entomology* (*Russell et al., 2013*)and *Mackenzie et al. (2012)*.

We defined a known vector-virus pair as one for which the full transmission cycle (i.e, infection of mosquito via an infected host (mammal or avian) or bloodmeal that is able to be transmitted via saliva) has been observed. Basing vector competence on isolation or intrathoracic injection bypasses several important barriers to transmission (*Hardy et al., 1983*), and may not be true evidence of a mosquito's ability to transmit an arbovirus. We found our definition to be more conservative than that which is commonly used in disease databases (e.g. Global Infectious Diseases and Epidemiology Network database), which often assumes isolation from wild-caught mosquitoes to be evidence of a mosquito's role as a vector. Therefore, a supplementary analysis investigates the robustness of our findings with regards to uncertainty in vector status by comparing the analysis reported in the main text to a second analysis in which any kind of evidence for association, including merely isolating the virus in wild-caught mosquitoes, is taken as a basis for connection in the virus-vector network (see Appendix 1 for analysis and results).

Fifteen mosquito traits (*Appendix 2—table 1*) and twelve virus traits (*Appendix 2—table 2*) were collected from the literature. For the mosquito species, the geographic range was defined as the number of countries in which the species has been collected, based on *Walter Reed Biosystematics Unit, (2016)*. While there are uncertainties in species' ranges due to false absences, this represents the most comprehensive, standardized dataset available that includes both rare and common mosquito species. A species' continental extent was recorded as a binary value of its presence by continent. A species' host range was defined as the number of taxonomic classes the species is known to feed on, with the Mammalia class further split into non-human primates and other mammals, because of the important role primates play in zoonotic spillovers of vector-borne disease (e.g. dengue, chikungunya, yellow fever, and Zika viruses) (*Weaver, 2005*; *Diallo et al., 2005*; *Weaver et al., 2016*). The total number of unique flaviviruses observed per mosquito species was calculated from our mosquito-flavivirus matrix. All other traits were based on consensus in the literature (see Appendix III for sources by species). For three traits – urban preference, endophily (a proclivity to bite indoors), and salinity tolerance – if evidence of that trait for a mosquito was not found in the literature, it was assumed to be negative.

We collected data on the following virus traits: host range (*Mahy, 2009*; *Mackenzie et al., 2012*; *Chambers and Monath, 2003*; *Cook and Zumla, 2009b*), disease severity (*Mackenzie et al., 2012*), human illness (*Chambers and Monath, 2003*; *Cook and Zumla, 2009*), the presence of a mutated envelope protein, which controls viral entry into cells (*Grard et al., 2010*), year of isolation (*Karabatsos, 1985*), and host range (*Karabatsos, 1985*). Disease severity was based on *Mackenzie et al. (2012)*, ranging from no known symptoms (e.g. Kunjin virus) to severe symptoms and significant human mortality (e.g. yellow fever virus). For each virus, vector range was calculated as the number of mosquito species for which the full transmission cycle has been observed. Genome length was calculated as the mean of all complete genome sequences listed for each flavivirus in the Virus Pathogen Database and Analysis Resource (http://www.viprbrc.org/). For more recently discovered flaviviruses not yet cataloged in the above databases (i.e., New Mapoon Virus, Iquape virus), viral traits were gathered from the primary literature (sources listed in Appendix 3).

### Predictive model

Following *Han et al. (2015)*, boosted regression trees (BRT) (*Friedman, 2001*) were used to fit a logistic-like predictive model relating the status of all possible virus-vector pairs (0: not associated, 1: associated) to a predictor matrix comprising the traits of the mosquito and virus traits in each

pair. Boosted regression trees circumvent many issues associated with traditional regression analysis (*Elith et al., 2008*), allowing for complex variable interactions, collinearity, non-linear relationships between covariates and response variables, and missing data. Additionally, this technique performs well in comparison with other logistic regression approaches (*Friedman, 2001*). Trained boosted regression tree models are dependent on the split between training and testing data, such that each model might predict slightly different propensity values. To address this, we trained an ensemble of 25 internally cross-validated BRT models on independent partitions of training and testing data. The resulting model demonstrated low variance in relative variable importance and overall model accuracy, suggesting models all converged to a similar result.

Prior to the analysis of each model, we randomly split the data into training (70%) and test (30%) sets while preserving the proportion of positive labels (known associations) in each of the training and test sets. Models were trained using the gbm package in *R* (*Ridgeway, 2015*), with the maximum number of trees set to 25,000, a learning rate of 0.001, and an interaction depth of 5. To correct for optimistic bias (*Smith et al., 2014*), we performed 10-fold cross validation and chose a bag fraction of 50% of the training data for each iteration of the model. We estimated the performance of each individual model with three metrics: Area Under the Receiver Operator Curve, specificity, and sensitivity. For specificity and sensitivity, which require a preset threshold, we thresholded predictions on the testing data based on the value which maximized the sum of the sensitivity and specificity, a threshold robust to the ratio of presence to background points in presence-only datasets (*Liu et al., 2016*). Variable importance was quantified by permutation (*Breiman, 2001*) to assess the relative contribution of virus and vector traits to the propensity for a virus and vector to form a pair. Because we transformed many categorical variables into binary variables (e.g., continental range as binary presence or absence by continent), the sum of the relative importance for each binary feature was summed to obtain a single value for the entire variable.

Each of our twenty-five trained models was then used to predict novel mosquito vectors of Zika by applying the trained model to a data set consisting of the virus traits of Zika paired with the traits of all mosquitoes for which flaviviruses have been isolated from wild caught individuals, and, depending on the species, may or may not have been tested in full transmission cycle experiments (a total of 180 mosquito species). This expanded dataset allowed us to predict over a large number of mosquito species, while reasonably limiting our dataset to those species suspected of transmitting flaviviruses. The output of this model was a propensity score ranging from 0 to 1. In our case, the final propensity score for each vector was the mean propensity score assigned by the twenty-five models. To label unobserved edges, we thresholded propensity scores at the value of lowest ranked known vector (*Liu et al., 2013*).

## Model validation

In addition to conventional performance metrics, we conducted additional analyses to further validate both this method of prediction, and our model specifically. To account for uncertainty in the vector-virus links in our initial matrix, we repeated our analysis for a vector-virus matrix with a less conservative definition of a positive link (field isolation and above), referred to as our supplementary model. Vector competence is a dynamic trait, and there exists significant intraspecific variation in the ability of a vector to transmit a virus for certain species of mosquitoes (*Diallo et al., 2005*; *Gubler et al., 1979*). Our supplementary model is based on a less conservative definition of vector competence and includes species implicated as vectors, but not yet verified through laboratory competence studies, and therefore accounts for additional uncertainty such as intraspecific variation.

While this approach is well-tested in epidemiological applications (*Parascandola, 2004*), it has only recently been applied to predict ecological associations, and, as such, has limitations unique to this application. To further evaluate this prediction method, we performed a modified 'leave-one-out' analysis, whereby we trained a model to a dataset from which a well-studied virus had been omitted, and then predicted vectors for this virus and compared them against a list of known vectors. We repeated this analysis for West Nile, dengue, and yellow fever viruses, following the same method of training as for our original model. While this analysis differs from our original method, it provides a more stringent evaluation of this method of prediction because the model is trained on an incomplete dataset and predicts on unfamiliar data, a more difficult task than that posed to our original model.

## Acknowledgements

The authors acknowledge Pasha Feinberg for assistance with data collection.

## Additional information

### Funding

| Funder | Grant reference number | Author |
|---|---|---|
| National Science Foundation | DEB-1640780 | Courtney C Murdock |
| University of Georgia | Presidential Fellowship | Michelle V Evans |
| National Institutes of Health | U01GM110744 | John M Drake |

The funders had no role in study design, data collection and interpretation, or the decision to submit the work for publication.

### Author contributions

MVE, Data curation, Formal analysis, Visualization, Writing—original draft, Writing—review and editing; TAD, Formal analysis, Methodology, Writing—original draft, Writing—review and editing; BAH, Resources, Formal analysis, Methodology, Writing—original draft, Writing—review and editing; CCM, Writing—original draft, Writing—review and editing; JMD, Conceptualization, Methodology, Writing—original draft, Writing—review and editing

### Author ORCIDs

Michelle V Evans, http://orcid.org/0000-0002-5628-0502
Tad A Dallas, http://orcid.org/0000-0003-3328-9958
Barbara A Han, http://orcid.org/0000-0002-9948-3078
John M Drake, http://orcid.org/0000-0003-4646-1235

## Additional files

### Major datasets

The following dataset was generated:

| Author(s) | Year | Dataset title | Dataset URL | Database, license, and accessibility information |
|---|---|---|---|---|
| Michelle V Evans, Tad A Dallas | 2017 | Data and Code to reproduce Evans et al 2017: "Data-driven identification of potential Zika virus vectors" | https://doi.org/10.6084/m9.figshare.4042488.v1 | Publicly accessible at figshare under a CC-BY licence ((https://figshare.com/). |

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

## Appendix 1

# Comparison model trained on virus isolation data

The primary model is trained on vector-virus pairs for which the full transmission cycle has been observed. However, many sources, such as the Global Infectious Diseases and Epidemiology Network database (GIDEON), interpret isolation of a virus in wild-caught mosquitoes as evidence of a mosquito's role as a vector. In order to investigate the robustness of our findings, we conducted a supplementary analysis in which any evidence for association, including isolation of the virus, is used as the basis for a link in the vector-virus network.

### Data collection

As in the primary model, the mosquito-virus pair matrix was constructed based on the Global Infectious Diseases and Epidemiology Network database (*GIDEON, 2016*), the International Catalog of Arboviruses Including Certain Other Viruses of Vertebrates (ArboCat) (*Karabatsos, 1985*), *The Encyclopedia of Medical and Veterinary Entomology* (*Russell et al., 2013*) and (*Mackenzie et al., 2012*). This resulted in a dataset containing 180 mosquito species and 37 viruses, for a total of 334 vector-virus pairs. The vector and virus trait datasets were identical to those used in the primary model (see Appendix 2 for lists of traits).

### Predictive model

We used boosted regression trees (*Friedman, 2001*) to fit a logistic-like predictive model relating the status of all possible virus-vector pairs (0: not associated, 1: associated) to a predictor matrix comprising the traits of the mosquito and virus traits in each pair. We fit a total of 25 models, applying different training and testing datasets to each, to reduce the dependence dependent on the split between training and testing data. Prior to the analysis of each model, we randomly split the data into training (70%) and test (30%) sets while preserving the proportion of positive labels in each of the training and test sets. Models were trained using the gbm package in *R* (*Ridgeway, 2015*), with the maximum number of trees set to 25,000 and a learning rate of 0.001. To correct for optimistic bias (*Smith et al., 2014*), we performed 10-fold cross validation and bagged 50% of the training data for each iteration of the model. These methods are identical to those used to train the primary model. We quantified variable importance by permutation (*Breiman, 2001*) to assess the relative contribution of virus and vector traits to the propensity for a virus and vector to form a pair. Each of our twenty-five trained models was then used to predict novel mosquito vectors of Zika over the whole virus-vector pair dataset, resulting in twenty-five propensity values assigned to each mosquito species, of which we took the mean. Our prediction dataset, therefore, consisted of the common virus traits of Zika paired with the common traits of all mosquitoes in our flavivirus dataset, for a total of 180 species. The output of this model was a propensity score ranging from 0 to 1. In our case, the final propensity score for each vector was the mean propensity score assigned by the twenty-five models. To label unobserved edges, we thresholded propensity at the value of lowest ranked known vector (*Liu et al., 2013*).

### Results

Boosted regression models trained on the weakest evidence of association accurately predicted mosquito vector-virus associations in the test dataset (AUC= 0.84 ±0.02). When thresholded at the value of the lowest ranked known vector, the model predicted 66 potential vectors of ZIKV, including 42 unknown vectors LABEL:table:predictions. The majority of predicted vectors were *Aedes* species (39 species), with *Culex* as the second most predicted genus (15

species). It included all but three of the vectors predicted by the main model (*Ae. occidentalis*, *Ru. frontosa*, *Cx. rubinotus*).

## Model comparisons

Our supplementary and primary models, trained on virus isolation and above and full transmission cycle, respectively, generally concur. The models are fairly correlated (Spearman's coefficient, $\rho$=0.508 when considering the propensities of all 180 species 1. However, when only comparing the correlation of propensities between those vectors above the threshold of lowest ranked known vector, the models become much more correlated ($\rho$=0.693). This suggests that our model has a higher sensitivity than specificity, and is better able to predict those vectors that are competent for ZIKV than those that are not. The predictive accuracy of our supplementary model was slightly lower than our primary model. However, this may be an indirect effect of a lower positive-negative label ratio in the dataset used in the primary model, which can artificially inflate AUC values (**Lobo et al., 2008**).

The models differ in their ability to differentiate between vectors and non-vectors. The distribution of propensities for our main model is more skewed towards lower propensity values than is the supplementary model 2. This is logical, as the dataset used to train the main model contains a higher proportion of zeros (e.g. vector-virus pairs with no known association) than the supplementary model. The difference in distributions is accounted for by a similar discrepancy in threshold propensity values based on the lowest ranked known vector. The main model, which has a higher frequency of near-zero propensities, uses a lower threshold value than the supplementary model, however both thresholds qualitatively lie above the majority of the distributions.

## Conclusion

In summary, our supplementary model predicts which mosquito species may test positive for ZIKV through isolation in wild-caught individuals. As isolation can be understood as evidence of a vector's role in transmission of a disease, our supplementary model may also be interpreted as a ranking of potential vectors of ZIKV, similar to our main model. In fact, both models are well correlated in their ranking of species, although the main model, which trains on fewer vector-virus links, predicts fewer vectors than the supplementary model. Those species predicted by both models, such as *Cx. quinquefasciatus* and *Ae. vexans*, should be prioritized for further research on their competency to transmit ZIKV. Furthermore, as suggested by the main model, the current geographic range at risk for ZIKV transmission in the United States should be expanded to include the range of these species ranked highly by both our main and supplementary models.

**Appendix 1—table 1.** Vector predictions by the supplementary model.

| Vector | GBM Prediction | SD |
|---|---|---|
| *Aedes aegypti* | 0.84 | 0.06 |
| *Aedes albopictus* | 0.81 | 0.07 |
| *Aedes vittatus* | 0.76 | 0.10 |
| *Aedes africanus* | 0.70 | 0.11 |
| *Aedes taylori* | 0.65 | 0.14 |
| *Aedes furcifer* | 0.65 | 0.14 |
| *Aedes luteocephalus* | 0.59 | 0.12 |
| *Aedes metallicus* | 0.59 | 0.13 |
| *Aedes opok* | 0.58 | 0.13 |
| *Culex quinquefasciatus* | 0.56 | 0.13 |

*Appendix 1—table 1 continued on next page*

*Appendix 1—table 1 continued*

| Vector | GBM Prediction | SD |
|--------|----------------|-----|
| Aedes tarsalis | 0.56 | 0.12 |
| Aedes scutellaris | 0.56 | 0.11 |
| Aedes minutus | 0.55 | 0.12 |
| Aedes polynesiensis | 0.53 | 0.11 |
| Mansonia uniformis | 0.52 | 0.12 |
| Aedes fowleri | 0.48 | 0.14 |
| Aedes vexans | 0.46 | 0.11 |
| Aedes dalzieli | 0.45 | 0.13 |
| Culex annulirostris | 0.45 | 0.08 |
| Mansonia africana | 0.42 | 0.12 |
| Psorophora ferox | 0.39 | 0.14 |
| Culex tarsalis | 0.38 | 0.09 |
| Culex tritaeniorhynchus | 0.37 | 0.08 |
| Culex pipiens | 0.37 | 0.13 |
| Culex neavei | 0.34 | 0.06 |
| Aedes vigilax | 0.34 | 0.07 |
| Aedes flavicollis | 0.33 | 0.14 |
| Aedes scapularis | 0.31 | 0.07 |
| Aedes taeniarostris | 0.31 | 0.13 |
| Aedes jamoti | 0.31 | 0.13 |
| Aedes circumluteolus | 0.30 | 0.13 |
| Eretmapodites inornatus | 0.30 | 0.15 |
| Aedes cumminsii | 0.29 | 0.11 |
| Culex vishnui | 0.28 | 0.05 |
| Aedes lineatopennis | 0.28 | 0.11 |
| Aedes neoafricanus | 0.27 | 0.11 |
| Aedes bromeliae | 0.26 | 0.10 |
| Culex guiarti | 0.26 | 0.06 |
| Culex perfuscus | 0.26 | 0.06 |
| Aedes stokesi | 0.26 | 0.12 |
| Culex telesilla | 0.25 | 0.06 |
| Anopheles gambiae | 0.24 | 0.11 |
| Sabethes chloropterus | 0.24 | 0.11 |
| Aedes hensilli | 0.24 | 0.09 |
| Aedes serratus | 0.23 | 0.06 |
| Aedes chemulpoensis | 0.23 | 0.08 |
| Aedes normanensis | 0.23 | 0.06 |
| Culex bitaeniorhynchus | 0.22 | 0.09 |
| Culex pseudovishnui | 0.22 | 0.05 |
| Aedes argenteopunctatus | 0.21 | 0.06 |
| Wyeomyia vanduzeei | 0.21 | 0.15 |
| Culex p. molestus | 0.21 | 0.06 |

*Appendix 1—table 1 continued*

| Vector | GBM Prediction | SD |
|---|---|---|
| *Culex salinarius* | 0.20 | 0.04 |
| *Aedes grahami* | 0.19 | 0.15 |
| *Anopheles coustani* | 0.19 | 0.08 |
| *Aedes longipalpis* | 0.18 | 0.18 |
| *Uranotaenia sapphirina* | 0.17 | 0.08 |
| *Aedes domesticus* | 0.17 | 0.06 |
| *Aedes abnormalis* | 0.17 | 0.06 |
| *Aedes natronius* | 0.17 | 0.06 |
| *Eretmapodites chrysogaster* | 0.17 | 0.08 |
| *Aedes mcintoshi* | 0.17 | 0.06 |
| *Aedes ochraceus* | 0.16 | 0.06 |
| *Culex fatigans* | 0.16 | 0.07 |
| *Anopheles amictus* | 0.16 | 0.06 |
| *Eretmapodites quinquevittatus* | 0.16 | 0.08 |

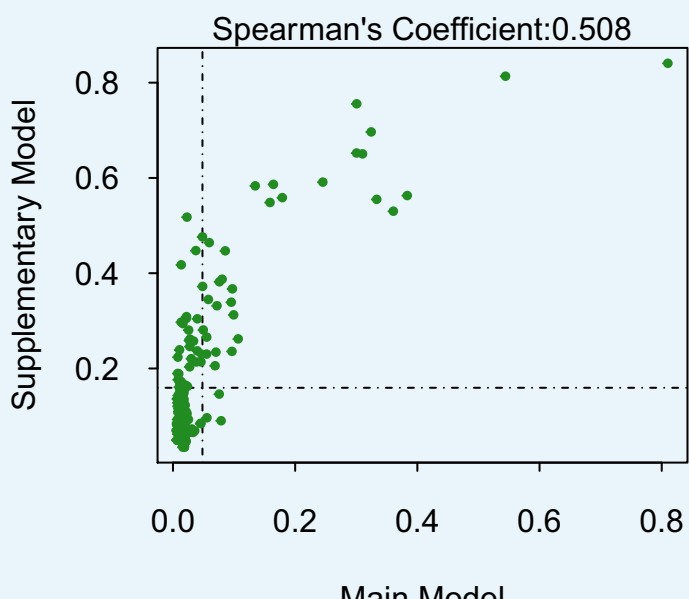

**Appendix 1—figure 1.** Propensity values of the main and supplementary models. Dashed lines represent corresponding threshold values for each model based on lowest ranked known vector propensities.

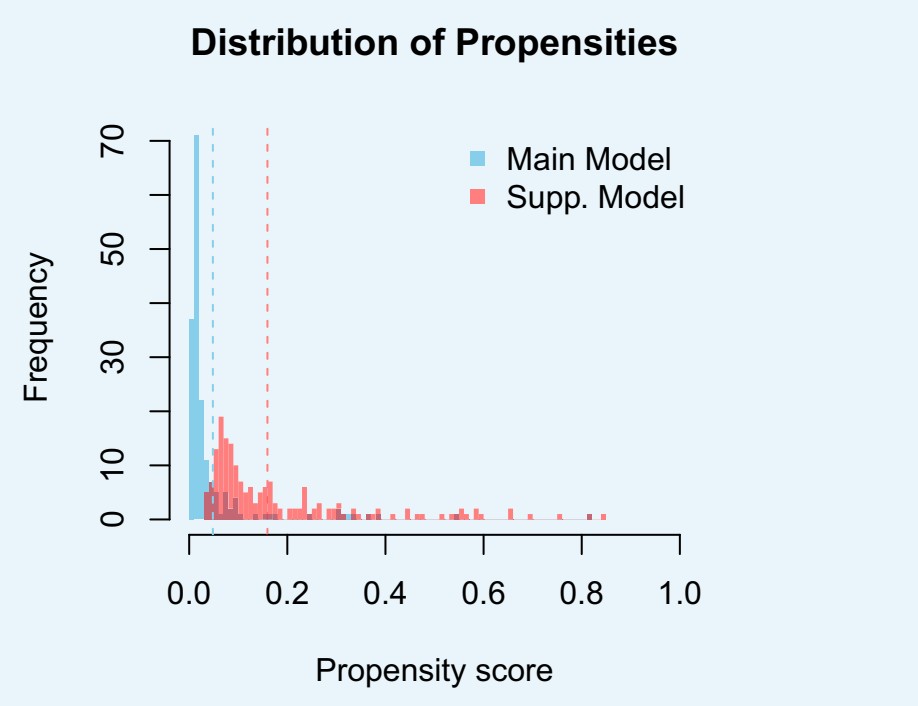

**Appendix 1—figure 2.** Distribution of propensity values for the main and supplementary models. Dashed lines represent corresponding threshold values for each model based on lowest ranked known vector propensities.

# Appendix 2

## Tables of vector and virus traits

**Appendix 2—table 1.** Table of mosquito traits used in model.

| Trait | Type | Subcategories |
|---|---|---|
| Anthropophily | binary | NA |
| Subgenus | factor | NA |
| Host breadth | numeric | NA |
| Host range | binary (x4) | Primate, Non-Primate Mammal, Bird, Cold-Blooded Vertebrate |
| Geographic area | numeric | NA |
| Continental range | binary (x8) | Africa, Middle East, Australia, Pacific, Asia, Europe, North America, South America |
| Biting time | binary (x4) | Dawn, Day, Dusk, Night |
| Artificia container breeder | binary | NA |
| Oviposition site | binary (x8) | Treehole, Natural Container, Permanent Fresh Water, Rockhole, Marsh, Swamp, Temporary Ground Pools, Rice Paddy |
| Habitat discrimination | numeric | NA |
| Salinity tolerance | binary | NA |
| Habitat permanence | binary | NA |
| Urban preference | binary | NA |
| Endophily | binary | NA |
| No. of flaviviruses vectored | numeric | NA |

**Appendix 2—table 2.** Table of virus traits used in model.

| Trait | Type | Subcategories |
|---|---|---|
| Group | factor | Japanese Encephalitis, Ntaya, Yellow Fever, Aroa, Dengue, Kokobera, Spondweni |
| Continental range | binary (x8) | Africa, Middle East, Australia, Pacific, Asia, Europe, North America, South America |
| Clade | factor | VI, VII, IX, X, XI, XII, XIV, |
| Year isolated | numeric | NA |
| Mutated envelope | binary | NA |
| Host breadth | numeric | NA |
| Host Range | binary (x6) | Human, Non-Human Primate, Rodent, Other Mammal, Bird, Marsupial |
| Mosquito vector breadth | numeric | NA |
| Vectored by other arthropods | binary | NA |
| Disease symptoms | binary (x2) | Encephalitis, Fever |
| Disease severity | numeric | NA |
| Genome length | numeric | NA |

## Appendix 3

### Primary sources used for vector and virus traits

**Appendix 3—table 1.** Primary sources for mosquito traits.

| Mosquito species | Sources |
|---|---|
| Aedeomyia africana | Robert et al. (1998), Harbach (2015), Omondi et al. (2015) |
| Aedeomyia catasticta | Harbach (2015), Jansen et al. (2009), Wright et al. (1981) |
| Aedes abnormalis | Iwuala (1981) |
| Aedes aegypti | Halstead (2008), Ramasamy et al. (2011) |
| Aedes africanus | Haddow (1961) |
| Aedes albopictus | Ramasamy et al. (2011) |
| Aedes alternans | NSW Health (2016), Russell et al. (2013), Knight et al. (2012) |
| Aedes argenteopunctatus | Harbach (2015), Fontenille et al. (1998) |
| Aedes bancroftianus | NSW Health (2016), Russell (1986), Harbach (2015) |
| Aedes bromeliae | Bennett et al. (2015), Beran (1994), Digoutte (1999) |
| Aedes caballus | Harbach (2015), Steyn and Schulz (1955) |
| Aedes canadensis | Carpenter and LaCasse (1974), Andreadis et al. (2004) |
| Aedes cantans | Renshaw et al. (1994, 1995), Service (1993) |
| Aedes cantator | Giberson et al. (2007) |
| Aedes chemulpoensis | Feng (1983) |
| Aedes cinereus | Morrison and Andreadis (1992), Anderson et al. (2007), Becker and Neumann (1983), Molaei et al. (2008) |
| Aedes circumluteolus | Jupp and McIntosh (1987), Paterson et al. (1964), Chandler et al. (1975) |
| Aedes cumminsi | Lane and Crosskey (2012) |
| Aedes curtipes | Harbach (2015), MacDonald et al. (1965), Knight and Hull (1953) |
| Aedes dalzieli | Fontenille et al. (1998) |
| Aedes domesticus | Harbach (2015), Lane and Crosskey (2012), Geoffroy (1987) |
| Aedes dorsalis | Aldemir et al. (2010), Wang et al. (2012) |
| Aedes flavicolis | Reinert (1970) |
| Aedes fluviatilis | Multini et al. (2015), Baton et al. (2013), Reinert et al. (2008) |
| Aedes fowleri | (Boussès et al., 2013) |
| Aedes furcifer | Beran (1994), Hopkins (1952) |
| Aedes grahami | Harbach (2015) |
| Aedes hensilli | Ledermann et al. (2014), Bohart and Ingram (1946) |
| Aedes ingrami | Lane and Crosskey (2012), Haddow (1946b, 1964, 1942) |
| Aedes jamoti | Harbach (2015), Le Berre and Hamon (1961) |
| Aedes japonicus | Kaufman and Fonseca (2014), Kampen and Werner (2014) |
| Aedes juppi | Harbach (2015), Jupp and Kemp (1998) |
| Aedes koreicus | Harbach (2015), Montarsi et al. (2013), Medlock et al. (2015) |
| Aedes lineatopennis | Harbach (2015), Amerasinghe and Indrajith (1995), Jupp (1967), Linthicum et al. (1985) |
| Aedes longipalpis | Harbach (2015) |
| Aedes luteocephalus | Diallo et al. (2012a), Service (1965b), Boorman (1961) |
| Aedes mcintoshi | Walter Reed Biosystematics Unit (2016), Harbach (2015) |
| Aedes mediolineatus | Harbach (2015) |

*Appendix 3—table 1 continued on next page*

*Appendix 3—table 1 continued*

| Mosquito species | Sources |
|---|---|
| *Aedes melanimon* | *Walter Reed Biosystematics Unit (2016), Barker et al. (2009), Chapman (1960)* |
| *Aedes metallicus* | *Harbach (2015), Beran (1994)* |
| *Aedes minutus* | *Harbach (2015), Diallo et al. (2012b)* |
| *Aedes natronius* | *Harbach (2015)* |
| *Aedes neoafricanus* | *Harbach (2015), Diallo et al. (2012b), Hervy et al. (1986)* |
| *Aedes normanensis* | *NSW Health (2016), Hearnden and Kay (1995)* |
| *Aedes notoscriptus* | *NSW Health (2016), Jansen et al. (2015), Nicholson et al. (2015), Derraik et al. (2007), Frances et al. (2002)* |
| *Aedes occidentalis* | *Harbach (2015), Evans (1926)* |
| *Aedes ochraceus* | *Corbet (1962), Lutomiah et al. (2014)* |
| *Aedes opok* | *Beran (1994), Herve et al. (1975), Germain et al. (1976)* |
| *Aedes polynesiensis* | *Young (2007)* |
| *Aedes procax* | *NSW Health (2016), Ryan and Kay (2000)* |
| *Aedes scapularis* | *Forattini et al. (1988)* |
| *Aedes scutellaris* | *Penn (1947)* |
| *Aedes serratus* | *Guimarães et al. (2000), Cardoso et al. (2010)* |
| *Aedes simulans* | *Harbach (2015)* |
| *Aedes sollicitans* | *Giberson et al. (2007), Carpenter and LaCasse (1974), Crans and Sprenger (1996), Crans et al. (1996)* |
| *Aedes stokesi* | *Harbach (2015), Reinert (1986)* |
| *Aedes taeniarostris* | *Eastwood et al. (2013)* |
| *Aedes tarsalis* | *Ellis et al. (2007)* |
| *Aedes taylori* | *Walter Reed Biosystematics Unit (2016)* |
| *Aedes togoi* | *Tsunoda et al. (2012), Lee and Hong (1995)* |
| *Aedes tremulus* | *Kay et al. (2000), Webb et al. (2016)* |
| *Aedes trivittatus* | *Carpenter and LaCasse (1974), Andreadis et al. (2004)* |
| *Aedes vexans* | *Boxmeyer and Palchick (1999), Aldemir et al. (2010)* |
| *Aedes vigilax* | *NSW Health (2016), Chapman et al. (1999)* |
| *Aedes vittatus* | *Boorman (1961), Selvaraj and Dwarakanath (1992)* |
| *Anopheles amictus* | *Hearnden and Kay (1995)* |
| *Anopheles barbirostris* | *Sriwichai et al. (2016), Amerasinghe and Indrajith (1995), Bashar et al. (2012)* |
| *Anopheles coustani* | *Fornadel et al. (2011), Mwangangi et al. (2013), Muriu et al. (2008), Mwangangi et al. (2007)* |
| *Anopheles crucians* | *Grieco et al. (2006), Qualls et al. (2012)* |
| *Anopheles domicola* | *Diagne et al. (1994)* |
| *Anopheles funestus* | *Gillies et al. (1968), Githeko et al. (1996)* |
| *Anopheles gambiae* | *Coggeshall (1944), Gillies et al. (1968), Huho et al. (2013)* |
| *Anopheles hyrcanus* | *Rueda et al. (2006, 2005), Ponçon et al., 2007), Aldemir et al. (2010)* |
| *Anopheles maculipennis* | *Aldemir et al. (2010), Brugman et al. (2015), Gordeev et al. (2005)* |
| *Anopheles meraukensis* | *Cooper et al. (2006), NSW Health (2016)* |
| *Anopheles paludis* | *Karch and Mouchet (1992), Mouchet (1957)* |
| *Anopheles pharoensis* | *Gillies et al. (1968), Taye et al. (2006)* |
| *Anopheles philippinensis* | *Toma et al. (2002), Silver (2007), Bashar et al. (2012)* |

*Appendix 3—table 1 continued*

| Mosquito species | Sources |
| --- | --- |
| Anopheles pretoriensis | Al-Sheik (2011), Shililu et al. (2003) |
| Anopheles punctipennis | Carpenter and LaCasse (1974) |
| Anopheles quadrimacula-tus | Carpenter and LaCasse (1974) |
| Anopheles subpictus | Sinka et al. (2011) |
| Anopheles tesselatus | Miyagi et al. (1983), Paramasivan et al. (2015) |
| Armigeres obturbans | Harbach (2015) |
| Coquillettidia aurites | Schwetz (1930), Njabo et al. (2009) |
| Coquillettidia linealis | Russell et al. (2013), Williams (2005), Webb et al. (2016) |
| Coquillettidia metallica | Njabo et al. (2009), Mcclelland ga et al. (1960) |
| Coquillettidia perturbans | Carpenter and LaCasse (1974), Anderson et al. (2007), Bosak et al. (2001), Callahan and Morris (1987) |
| Coquillettidia richiardii | Ventim et al. (2012), Serandour et al. (2006), Versteirt et al. (2013) |
| Coquillettidia venezuelen-sis | Guimarães et al. (2000), Degallier et al. (1978) |
| Culex adamesi | Sirivanakarn and Galindo (1980) |
| Culex annulirostris | NSW Health (2016), Hall-Mendelin et al. (2012), Williams and Kokkinn (2005) |
| Culex antennatus | Gad et al. (1995), Karch et al. (1993), Morsy et al. (1990), Kenawy et al. (1998) |
| Culex australicus | NSW Health (2016), Russell (2012) |
| Culex bahamensis | Lopes (1997) |
| Culex bitaeniorhynchus | Kulkarni and Rajput (1988), Fakoorziba and Vijayan (2008), Har-bach (1988) |
| Culex caudelli | Alfonzo et al. (2005), Chadee and Tikasingh (1989) |
| Culex coronator | Yee and Skiff (2014), de Oliveria et al. (1985) |
| Culex crybda | de Oliveria et al. (1985) |
| Culex duttoni | Mwangangi et al. (2009) |
| Culex epidesmus | Kanojia (2003), Reisen et al. (1976) |
| Culex fatigans | Flordia Medical Entomology Laboratory (2016), Liu et al. (1960), Robinson (2005) |
| Culex fuscocephala | Ohba et al. (2015), Kulkarni and Rajput (1988), Amerasinghe and Munasingha (1994), Wang (1975) |
| Culex gelidus | Williams (2005), Sudeep (2014) |
| Culex guiarti | Logan et al. (1991) |
| Culex modestus | Veronesi et al. (2012), Radrova et al. (2013), Muñoz et al. (2012), Chalvet-Monfray et al. (2007), Fyodorova et al. (2006) |
| Culex nakuruensis | Someren (1967) |
| Culex neavei | Diallo et al. (2012a), Nikolay et al. (2012), Fall et al. (2013, 2011) |
| Culex nebulosus | Adebote et al. (2006), Okorie (1978), Davis and Philip (1931) |
| Culex nigripalpus | Laporta et al. (2008), Carpenter and LaCasse (1974), Flordia Medical Entomology Laboratory. (2016) |
| Culex p. molestus | Robinson (2005), Gomes et al. (2013) |
| Culex perexiguus | Muñoz et al. (2012), Ammar et al. (2012) |
| Culex perfuscus | Hopkins (1952), Diallo et al. (2014), Service (1993) |
| Culex pipiens | Harbach (1988), Anderson et al. (2007) |
| Culex poicilipes | Muturi et al. (2008), Yamar et al. (2005), Chevalier et al. (2004) |

*Appendix 3—table 1 continued on next page*

Appendix 3—table 1 continued

| Mosquito species | Sources |
|---|---|
| Culex pruina | Wanson and Lebred (1946) |
| Culex pseudovishnui | Fakoorziba and Vijayan (2008), Reisen et al. (1976), Amerasinghe and Indrajith (1995), Reuben et al. (1992) |
| Culex pullus | Johansen et al. (2009), Webb et al. (2016) |
| Culex quinquefasciatus | Flordia Medical Entomology Laboratory (2016), DeGroote and Sugumaran (2012) |
| Culex restuans | Apperson et al. (2002), Ebel et al. (2005), Kilpatrick et al. (2005), Molaei et al. (2008) |
| Culex rubinotus | Jupp et al. (1976) |
| Culex salinarius | Rochlin et al. (2008), Mackay et al. (2010), Rey et al. (2006) |
| Culex sitiens | NSW Health (2016), Prummongkol et al. (2012) |
| Culex spissipes | Takahashi (1968), Degallier et al. (1978) |
| Culex squamoses | NSW Health (2016), Jansen et al. (2009) |
| Culex taeniopus | Davies (1978), 1975), Lopes (1996) |
| Culex tarsalis | Reisen (1993), Rueger et al. (1964) |
| Culex telesilla | Njogu and Kinoti (1971) |
| Culex thalassius | Kerr (1932), Snow and Boreham (1978), Service (1993), Kirby et al. (2008) |
| Culex theileri | Aldemir et al. (2010), Muñoz et al. (2012), Simsek (2004) |
| Culex tritaeniorhynchus | Kanojia and Geevarghese (2004), Fakoorziba and Vijayan (2008), Flemings (1959), Amerasinghe and Munasingha (1994), Mwandawiro et al. (1999), Bhattacharyya et al. (1994), Reuben (1971) |
| Culex univittatus | Jupp (1967), Chandler et al. (1975), Jupp and Brown (1967) |
| Culex virgultus | Carpenter and LaCasse (1974) |
| Culex vishnui | Chen et al. (2014), Bhattacharyya et al. (1994), Ohba et al. (2015) |
| Culex vomerifer | Ferro et al. (2003), Natal et al. (1998), Suárez-Mutis et al. (2009), Sallum and Forattini (1996) |
| Culex weschei | Snow and Boreham (1973), Lane and Crosskey (2012) |
| Culex whitmorei | Begum et al. (1986), Reisen et al. (1976), Peiris et al. (1992) |
| Culex zombaensis | Lane and Crosskey (2012), Logan et al. (1991) |
| Culiseta alaskensis | Frohne (1953) |
| Culiseta impatiens | Sommerman (1964), Frohne (1953), Murdock et al. (2010), Smith (1966) |
| Culiseta inornata | Carpenter and LaCasse (1974), Smith (1966), Belton (1979) |
| Culiseta melanura | Molaei et al. (2006), Mahmood and Crans (1998), Flordia Medical Entomology Laboratory (2016), Hickman and Brown (2013) |
| Deinocerites pseudes | Martin et al. (1973), Peyton et al. (1964) |
| Eretmapodites chrysogaster | Doucet and Cachan (1961), Sylla et al. (2013), Service (1965a), Haddow (1946b) |
| Eretmapodites inornatus | Haddow (1946a) |
| Eretmapodites oedipodeios (oedipodius) | Haddow (1946a), de Cunha Ramos and Ribeiro (1990) |
| Eretmapodites quinquevittatus | Bohart and Ingram (1946), Jupp and Kemp (2002), Lounibos (1980) |
| Eretmapodites silvestris | Lounibos (1980), Hoogstraal and Knight (1951) |
| Ficalbia flavens | King and Hoogstraal (1946) |
| Haemagogus anastasionis | Van der Kuyp (1949), Bueno-MarÃ etal. (2015), Maestre-Serrano et al. (2013) |

Appendix 3—table 1 continued on next page

*Appendix 3—table 1 continued*

| Mosquito species | Sources |
|---|---|
| Haemagogus celeste | Bueno-MarÃ etal., 2015, Maestre-Serrano et al. (2013), Beran (1994), Chadee et al. (1985) |
| Haemagogus equinus | Chadee et al. (1985, 1993), Waddell and Taylor (1945) |
| Haemagogus janthinomys | Arnell (1973), Alencar et al. (2005), Chadee et al. (1992) |
| Haemagogus leucocelaenus | Alencar et al. (2008), Pinto et al. (2009) |
| Haemagogus spegazzinii | Arnell (1973), Galindo et al. (1951, 1950) |
| Mansonia africana | Karch et al. (1993), Chandler et al. (1975), Hopkins (1952) |
| Mansonia septempunctata | NSW Health (2016), Harbach (2015) |
| Mansonia titillans | Carpenter and LaCasse (1974), Viana et al. (2010), Stein et al. (2013) |
| Mansonia uniformis | Sabesan et al. (1991), Kumar et al. (1989), Wharton (1962) |
| Mimomyia hispida | Boreham et al. (1975), Harbach (2015) |
| Mimomyia lacustris | Harbach (2015) |
| Mimomyia splendens | Boreham et al. (1975), Robert et al. (1998) |
| Orthopodomyia signifera | Hanson et al. (1995), Burkett-Cadena (2013) |
| Psorophora albipes | Alfonzo et al. (2005), dos Santos Silva et al. (2012), Guimarães et al. (2000) |
| Psorophora columbiae | Carpenter and LaCasse (1974) |
| Psorophora ferox | Carpenter and LaCasse (1974), Flordia Medical Entomology Laboratory (2016), Degallier et al. (1978), Molaei et al. (2008) |
| Runchomyia frontosa | Cardoso et al. (2015), Heinemann et al. (1980) |
| Sabethes albiprivus | Gomes et al. (2010), Pedro et al. (2008) |
| Sabethes belisarioi | Pinto et al. (2009) |
| Sabethes chloropterus | Beran (1994), Pinto et al. (2009), Galindo (1958) |
| Sabethes soperi | Navarro et al. (2015), Harbach (2015) |
| Uranotaenia mashonaensis | Harbach and Schnur (2007) |
| Uranotaenia sapphirina | Cupp et al. (2003), Crans (2016) |
| Uranotaenia unguiculata | Khoshdel-Nezamiha et al. (2014), Ramsdale and Snow (2001), Sebesta et al. (2010), Bagirov et al. (1994), Kenawy et al. (1987) |

**Appendix 3—table 2.** Primary sources for virus traits.

| Virus | Sources |
|---|---|
| Alfuy Virus | Mackenzie et al. (2012) |
| Bagaza virus | Mahy (2009), Llorente et al. (2015), Gamino et al. (2012) |
| Banzi virus | Grard et al. (2010), Karabatsos (1985) |
| Bouboui virus | Grard et al. (2010), Cook and Zumla (2009) |
| Bussuquara virus | Beran (1994) |
| Dengue type 1 | Cook and Zumla (2009) |
| Dengue type 2 | Cook and Zumla (2009) |
| Dengue type 3 | Cook and Zumla (2009) |
| Dengue type 4 | Cook and Zumla (2009) |
| Edge Hill virus | Mackenzie et al. (2012), Doherty et al. (1964) |
| Iguape Virus | Coimbra et al. (1993), Mahy (2009) |
| Ilheus virus | Mahy (2009), Chambers and Monath (2003), Laemmert and Hughes (1947), Aitken and Anderson (1959) |

*Appendix 3—table 2 continued*

| Virus | Sources |
|---|---|
| Israel turkey meningoence-phalomyelitis virus | *Mahy (2009), Nir (1972)* |
| Japanese encephalitis virus | *Mahy (2009), Burke and Leake (1988), Gresser et al. (1958)* |
| Jugra virus | None |
| Kedougou virus | *Cook and Zumla (2009), Diagne et al. (2015a)* |
| Kokobera virus | *Cook and Zumla (2009), Lequime and Lambrechts (2014)* |
| Koutango virus | *Chambers and Monath (2003), Cook and Zumla (2009)* |
| Kunjin virus | *Mahy (2009), Mackenzie et al. (2012)* |
| Murray Valley encephalitis virus | *Cook and Zumla (2009), Mackenzie et al. (2012)* |
| Naranjal virus | *Mahy (2009)* |
| New Mapoon virus | *Nisbet et al. (2005), Mahy (2009)* |
| Ntaya virus | *Mahy (2009)* |
| Rocio virus | *Mahy (2009), Cook and Zumla (2009)* |
| Saboya virus | *Mahy (2009), Traoré-Lamizana et al. (2001)* |
| Sepik virus | *Mackenzie et al. (2012), Cook and Zumla (2009)* |
| Spondweni virus | *Chambers and Monath (2003), Cook and Zumla (2009)* |
| St. Louis encephalitis virus | *Mackenzie et al. (2012), Cook and Zumla (2009)* |
| Stratford virus | *Mackenzie et al. (2012)* |
| Tembusu virus | *Mahy (2009), Tang et al. (2015)* |
| Uganda S virus | *Mahy (2009)* |
| Usutu virus | *Mahy (2009), Chambers and Monath (2003), Cook and Zumla (2009)* |
| Wesselbron | *Mahy (2009), Chambers and Monath (2003), Cook and Zumla (2009)* |
| West Nile virus | *Mackenzie et al. (2012), Cook and Zumla (2009a), Mores et al. (2007), Turell et al. (2001)* |
| Yaounde virus | *Mackenzie et al. (2012)* |
| Yellow fever virus | *Mahy (2009)* |
| Zika virus | *Chambers and Monath (2003), Cook and Zumla (2009)* |

