## [Decision Letter]

Thank you for submitting your article "Data-driven identification of potential Zika virus vectors" for consideration by *eLife*. Your article has been reviewed by three peer reviewers and the evaluation has been overseen by a Reviewing Editor and Prabhat Jha as the Senior Editor. The following individual involved in review of your submission has agreed to reveal his identity: Anthony Wilson (Reviewer #3).

The reviewers have discussed the reviews with one another and the Reviewing Editor has drafted this decision to help you prepare a revised submission.

Summary:

Overall, the feedback from this round of reviews was positive and all reviewers agreed that this is a novel approach to the identification of Zika's mosquito vectors. However, following a discussion among all three reviewing editors, some key themes emerged that we would like to see addressed before making a final decision on this manuscript. While we would like to see you address all of the major and minor comments of the reviewers, in particular we would like to see you pay particular attention to:

1) More rigorous model evaluation- from a statistical point of view, yes the model AUCs seem reasonable, but we all felt that this alone, was not enough to judge how effective these predictions were. A key assumption of the approach is that vector competence is associated with the kinds of traits that are readily observable and comparable. In addition to this, many key indicators that might be important for this are not included/considered, e.g. sub-species of vectors, evolutionary changes in vector competence. Judging by the top model predictors, it seems that there are very few traits that offer much explanatory power in this respect and perhaps a more nuanced explanation of why the model predict each of these new Zika vector species is required. Additionally we would like to see validation against a known outcome, e.g. if the vector-virus pairs of dengue, yellow fever or West Nile were left out of the fitting set what vector species would be predicted for each of these?

2) The caveats of this particular study need to be more appropriately articulated. The main output of this model is a list of candidate vector species to be tested experimentally for Zika virus competence (and ideally prioritised by likelihood of competence as predicted by the model and public health importance- i.e. additional population at risk, none of which is currently done in the paper). As it has been demonstrated many times that vector competence does not equate to sufficient vectorial capacity to cause outbreaks, let alone sustain transmission, it is perhaps premature to suggest that new risk assessments need to include other vector species at this stage. Please revise this and consider refocusing on providing useful recommendations for follow up studies.

Essential revisions:

Reviewer #1:

Evans and colleagues have submitted a manuscript detailing a novel and well conducted analysis that addresses a timely question of international public health importance. While, at times, the manuscript does overstate the significance of the findings and omits some important limitations, it is fundamentally an exciting study that could be of interest to a broad range of readers.

I am struggling to reconcile the parallel findings of high model performance (as demonstrated by AUC) and the finding that subgenus and continental range (two variables with extremely limited degrees of freedom, especially where Zika is concerned) contribute a high amount of the models' power. It seems that these very general and non-specific covariates would naturally lead to low specificity- how does the model AUC vary when predicting vector virus pairs for viruses with different characteristics – are very limited geographic scope viruses much easier to predict? If so, is there a more appropriate model evaluation measure for a broadly distributed disease such as Zika (and thus differing thresholds)? It would at least be useful to explain what features of Zika lend it to such a high number of predicted vectors relative to, say dengue.

This analysis quite rightly restricts its predictions to binary endpoint of vector competence. There is, however, a big difference between a competent vector and a vector that presents a true epidemiological risk. This is exemplified by *Ae. aegypti* vs *Ae. albopictus* in dengue with the former being responsible for the vast majority of transmission (see Lambrechts et al. 2010 PLoS NTDs). While I agree with the authors that these findings warrant further investigation of the competence of these species, to suggest that they need to be included in Zika risk maps makes too many assumptions (e.g. epidemiological significance, that other species will not outcompete their role as a vector, etc.) that are not supported by analysis in this manuscript. I would suggest re-wording to reduce the emphasis on this suggestion and including more limitations on why these vector species may not ultimately be epidemiologically significant.

Reviewer #2:

This study used data-driven, machine-learning algorithms to identify potential vectors of Zika virus. Although the premise of the study is good, several shortcomings of the approach make it difficult to ascertain the relevance of the results, which could be largely misleading. Some of the most important shortcomings are listed below.

The authors set out to "address the problem of identifying potential vector candidates in a suitable time frame" because "the amount of time required for analyses can delay decision making". However, identifying candidate Zika virus vectors would not preclude their subsequent empirical validation. So it is unclear what is the applied value of this modeling exercise.

A strong, unjustified assumption underlying the approach is that "the propensity of mosquito species to associate with Zika virus is statistically associated with common mosquito traits".

Documented implication of a given mosquito species in the transmission cycle of an arbovirus does not necessarily imply the universal importance of a vector species in the transmission cycle of this arbovirus. Vectorial capacity results from the combination of several factors, so that in certain local conditions (e.g., high mosquito population density, high temperature) even a poorly competent vector could play a significant role in transmission.

Vector status of a mosquito species for a given arbovirus cannot be permanently defined. Vector status is a dynamic process that can rapidly evolve (e.g., recent adaptation of chikungunya virus to *Aedes albopictus*).

Intra-species variation, which can be substantial for several mosquito traits, was ignored.

There was no empirical validation of the modeling approach.

Reviewer #3:

Vector-borne pathogens are emerging with increased frequency. This manuscript presents an interesting and potentially useful approach to the incrimination of vectors of novel emerging pathogens and I believe it to be worthy of publication. I have a couple of queries and suggestions, detailed below.

Results section, third paragraph: I suspect one criticism of this approach will be that it results in overly broad predictions (although even if true that would not mean it will not be useful in suggesting targets for epidemiological study). I would like the authors to add the result of applying the same method to 'predict' the vectors of dengue virus, yellow fever and West Nile virus when information on the respective viruses is removed from the training data.

Discussion section, first paragraph: could the authors elaborate on the additional regions or populations at risk from transmission based on their expanded 'worst case' vector list – or, even better, which newly-incriminated vectors make most difference to this? This should be possible using the dataset they have already collected (with the caveat that this is a worst-case scenario, and vector density and local environmental conditions also affect the potential for transmission).

---

## [Author Response]

*Summary:*

*Overall, the feedback from this round of reviews was positive and all reviewers agreed that this is a novel approach to the identification of Zika's mosquito vectors. However, following a discussion among all three reviewing editors, some key themes emerged that we would like to see addressed before making a final decision on this manuscript. While we would like to see you address all of the major and minor comments of the reviewers, in particular we would like to see you pay particular attention to:*

*1) More rigorous model evaluation- from a statistical point of view, yes the model AUCs seem reasonable, but we all felt that this alone, was not enough to judge how effective these predictions were. A key assumption of the approach is that vector competence is associated with the kinds of traits that are readily observable and comparable. In addition to this, many key indicators that might be important for this are not included/considered, e.g. sub-species of vectors, evolutionary changes in vector competence. Judging by the top model predictors, it seems that there are very few traits that offer much explanatory power in this respect and perhaps a more nuanced explanation of why the model predict each of these new Zika vector species is required. Additionally we would like to see validation against a known outcome, e.g. if the vector-virus pairs of dengue, yellow fever or West Nile were left out of the fitting set what vector species would be predicted for each of these?*

Thank you for this comment. To clarify, the association of traits with competence is not an “assumption” of the approach, but rather a conjecture in the form of a “working hypothesis” that is supported or not by the analysis. (Our original language was imprecise in this respect, and we have corrected it.) While we agree that sub-species identity or evolutionary change could be important, they are not features that are consistently available across this set of mosquitoes and cannot be readily mapped even for those species for which they are known. Thus, although they might be scientifically important for further research, they are not practically useful for our purposes. The limitation that this creates is that our model may not be as predictive as it could be with other information sources. Since new information would not erode the predictive performance of our model, we believe that the model is “at least as good” as measured, although possibly it can be improved in the future. Nonetheless, we agree that “more rigorous model evaluation” is warranted and thank the reviewers for their suggestions in this respect. The new analyses we performed in response to reviewer comments, as described below, have now further validated our technique and increased our confidence in the models’ predictions.

Particularly, we followed the suggestion of reviewers to validate our method by predicting the vectors of several of the “better-studied” viruses. Thus, we fit models to datasets iteratively omitting West Nile virus, yellow fever virus, and dengue virus to compare the models’ predictions against the “known outcome” of vectors of these viruses (subsections “Model Validation”). We note that this is not a perfect validation. First, there may yet be unidentified vectors of, say, West Nile virus, that if predicted by our model would score as “false positive” when in fact correct. Further, omitting data in model training and subsequently predicting onto the omitted data is a more difficult task than that asked of our original model (in which it at least had access to a portion of Zika virus links, i.e., the known ones). Nonetheless, these models were able to predict vectors of these viruses remarkably well for West Nile virus and dengue virus, although not for yellow fever virus (which we think informative). Specifically, yellow fever virus was special among the “better- studied” viruses in that the majority of its vectors were linked only with yellow fever virus. Thus, it would not be expected that other virus-mosquito pairs would be very informative. Omitting these links from the training dataset, therefore, resulted in incorrect predictions of these links (AUC = 0.47). This was especially true for species of the *Haemogogus* genus, which was completely omitted from the training dataset. In contrast, the vectors of West Nile virus and Dengue virus were highly predictable with models fit to the partial data (with AUC of 0.693 and 0.918, respectively). Since this was the performance on a more difficult task than our original model (i.e., no data from the target viral species rather than partial data from the target species), we think these quantities give roughly a lower bound – what we might call a worst case scenario – since it is a lower bound in the sense that the test data cannot be verified to be complete and since the task is harder than the one we attempt for Zika. (Of course, we advise against over-interpretation here – we have only two species which have highly divergent AUC scores, although both are significantly greater than random guessing.

In further response to the point about covariates, we note that data-driven modeling differs from traditional statistical modeling in that the relationship between predictor and response variables is not predetermined or explicitly stated (Breiman 2001, for instance, discusses this issue at length). Inferring causal relationships between predictor and response variables, therefore, is not advised. While we can describe trends within our dataset from our analysis, concluding the mechanism responsible for predicting a positive link, and attempting to generalize from this, would be conjecture. For this reason, we do not go into detail on why these vectors were ranked highly by the model.

*2) The caveats of this particular study need to be more appropriately articulated. The main output of this model is a list of candidate vector species to be tested experimentally for Zika virus competence (and ideally prioritised by likelihood of competence as predicted by the model and public health importance- i.e. additional population at risk, none of which is currently done in the paper). As it has been demonstrated many times that vector competence does not equate to sufficient vectorial capacity to cause outbreaks, let alone sustain transmission, it is perhaps premature to suggest that new risk assessments need to include other vector species at this stage. Please revise this and consider refocusing on providing useful recommendations for follow up studies.*

We agree that the results of this study should not be misinterpreted as a direct indicator of disease transmission risk. Indeed, the main motivation for limiting our discussion of transmission risk is to avoid over-interpretation of model results, which identify candidate species for empirical testing of vector competence. Of course, it does not follow that a competent species will also have high vectorial capacity, but a species with high vectorial capacity must be competent. We do, however, mention potential consequences (e.g., expanded range of the disease, difficulties in vector control) if several of these species were to be found competent to transmit Zika virus, and to illustrate the importance of validating our predictions. For specific examples, see Discussion section. We are puzzled by the statement that it may be “premature to suggest that new risk assessments need to include other vector species at this stage”. Of course, we do not know whether the additional species identified here provide sufficient vectorial capacity to cause outbreaks or sustain transmission. Therefore, we agree it would be premature to infer that testing for vector competence would

improve our assessment of transmission risk, but a first step towards this ultimate goal is to assess whether particular species are competent, so that they may be prioritized for the experimental testing required to confirm vectorial capacity (and to quantify transmission risk). We clarify these caveats (i.e., competence does not equal vectorial capacity or transmission risk) in the Discussion section.

*Reviewer #1:*

*Evans and colleagues have submitted a manuscript detailing a novel and well conducted analysis that addresses a timely question of international public health importance. While, at times, the manuscript does overstate the significance of the findings and omits some important limitations, it is fundamentally an exciting study that could be of interest to a broad range of readers.*

*I am struggling to reconcile the parallel findings of high model performance (as demonstrated by AUC) and the finding that subgenus and continental range (two variables with extremely limited degrees of freedom, especially where Zika is concerned) contribute a high amount of the models' power. It seems that these very general and non-specific covariates would naturally lead to low specificity- how does the model AUC vary when predicting vector virus pairs for viruses with different characteristics – are very limited geographic scope viruses much easier to predict? If so, is there a more appropriate model evaluation measure for a broadly distributed disease such as Zika (and thus differing thresholds)? It would at least be useful to explain what features of Zika lend it to such a high number of predicted vectors relative to, say dengue.*

We are not sure what the Reviewer #1 means by “degrees of freedom” in this context – possibly the amount of variation in the input variable, which is how we have interpreted it in our response. In this case, we note that there is an important distinction between our reported performance (AUC, which represents classification accuracy of our model and may be calculated virus by virus) versus variable importance, which represents how important a particular variable is for classification accuracy, and is based on the sensitivity of an AUC calculated for all vector-virus links in the testing dataset, not only those of Zika. For the former, the results of the added”leave-one out” analysis offers some insight into how each vector-virus link impacts model performance (reported above). In that case, we did test model performance against a known outcome, and found similarly high model performance. (subsections “Model Validation”). We hesitate to consider “limited geographic scope viruses” vs. other viruses. For many decades, Zika and West Nile virus would both have been considered to have limited geographic scope, though now are among the most widespread flaviviruses. This suggests that geographic range may not be as static a trait of viruses, particularly those that are emerging. In contrast,”subgenus” is assigned on the basis of phylogeny and therefore more intrinsic and less sensitive to temporal expansion.

*This analysis quite rightly restricts its predictions to binary endpoint of vector competence. There is, however, a big difference between a competent vector and a vector that presents a true epidemiological risk. This is exemplified by Ae. aegypti vs Ae. albopictus in dengue with the former being responsible for the vast majority of transmission (see Lambrechts et al. 2010 PLoS NTDs). While I agree with the authors that these findings warrant further investigation of the competence of these species, to suggest that they need to be included in Zika risk maps makes too many assumptions (e.g. epidemiological significance, that other species will not outcompete their role as a vector, etc.) that are not supported by analysis in this manuscript. I would suggest re-wording to reduce the emphasis on this suggestion and including more limitations on why these vector species may not ultimately be epidemiologically significant.*

We entirely agree with the reviewer that these species may not ultimately be epidemiologically significant, as vector competence is only one factor that influences overall risk. We have modified our language throughout to further clarify that vector competence is only one feature that contributes to the overall importance of a vector, specifically mentioning biological traits that may limit a species role in transmission. Additionally, we clarify that the primary aim of this modeling exercise is to inform empirical validation of Zika vectors. We nonetheless think it reasonable to show that the distribution of species in the US that are plausible vectors is much larger than risk maps currently widely circulated. Just as the reviewer would not want readers to infer that we think a high level of transmission is likely throughout this range, we are dismayed that risk maps currently in circulation do not better state their objectives – particularly, they are not meant to imply that it is impossible to acquire Zika virus outside the mapped region

*Reviewer #2:*

*This study used data-driven, machine-learning algorithms to identify potential vectors of Zika virus. Although the premise of the study is good, several shortcomings of the approach make it difficult to ascertain the relevance of the results, which could be largely misleading. Some of the most important shortcomings are listed below.*

*The authors set out to "address the problem of identifying potential vector candidates in a suitable time frame" because "the amount of time required for analyses can delay decision making". However, identifying candidate Zika virus vectors would not preclude their subsequent empirical validation. So it is unclear what is the applied value of this modeling exercise.*

We strongly agree that empirical validation of these vectors is necessary to better predict risk of Zika transmission, and believe the value of our exercise is that it provides a list of candidate vectors that should be prioritized by those validating vector competence. We have edited our manuscript to clarify that our results are not meant to replace empirical validation of vectors, but to guide empirical work and focus initial vector competence studies on these candidate species when the results are time-sensitive.

*A strong, unjustified assumption underlying the approach is that "the propensity of mosquito species to associate with Zika virus is statistically associated with common mosquito traits".*

Our initial phrasing of this assertion was misleading. Our study is not predicated on the assumption of this relationship, but, rather provides a test of this conjecture. Our model’s ability to draw on vector and virus traits associated with positive links, and use this information to accurately predict vectors of flaviviruses (as evidenced by its performance on the testing data) supports the theory that a vector’s propensity to associate with a specific flaviviruses is statistically associated with measurable traits.

*Documented implication of a given mosquito species in the transmission cycle of an arbovirus does not necessarily imply the universal importance of a vector species in the transmission cycle of this arbovirus. Vectorial capacity results from the combination of several factors, so that in certain local conditions (e.g., high mosquito population density, high temperature) even a poorly competent vector could play a significant role in transmission.*

We entirely agree. We make no claims of “universal importance”. Indeed, we have endeavored throughout not to overstate the implications of our study. While our model can predict probable competency, it does not speak to the relative vectorial capacity of predicted species. We have attempted to be precise throughout, specifically in Discussion paragraph four, and would welcome the reviewer to identify any further instances where we overstep.

*Vector status of a mosquito species for a given arbovirus cannot be permanently defined. Vector status is a dynamic process that can rapidly evolve (e.g., recent adaptation of chikungunya virus to Aedes albopictus).*

Of course, we agree. The fact of biological evolution means all biological associations are necessarily malleable. What we think is interesting and important is that species-level associations are stable enough to enable predictions to be made – as documented here. Nonetheless, we appreciate the reviewer’s counter-example. To incorporate vector status uncertainty in our model, we created a supplementary model trained on a dataset whose definition of a vector is less conservative, thereby including those species whose vector status is more uncertain. Predictions of vectors between the two models were correlated, although the model’s performance was lower, illustrating that additional uncertainty does lower model performance, but without significantly altering qualitative results of the model. We added additional discussion about this uncertainty and how we incorporated this into the analysis.

*Intra-species variation, which can be substantial for several mosquito traits, was ignored.*

Yes, comparative studies provide insight across species rather than across populations. Given that many of the biological and behavioral traits of mosquitoes and viruses vary intraspecifically, we attempted to address uncertainty as much as possible in the analysis, and remain transparent in cases where we could not. The inability to account for uncertainty in predictor variables at the species level is a limitation of all maps and dynamical models we are aware of. The development of methods to propagate uncertainty in predictor variables to the response variable is an important field of research and would improve the accuracy of models such as this. We note that within-species variation is understudied, and our manuscript accordingly calls for additional studies. (Discussion paragraph seven)

*There was no empirical validation of the modeling approach.*

While we did statistically validate our predictions by evaluating the model’s performance on testing data, empirical validation (i.e., experimental testing of mosquito species for Zika competence) is outside the purview of this study. The objective of this study was to create a list of potential mosquito vectors of Zika virus that should then be validated through empirical studies of vector

competence. A list of these likely vectors provides a starting point for empirical studies to support more efficient use of limited time and resources. We underscore the utility of our results for prioritizing necessary empirical validation in the lab and the field.

*Reviewer #3:*

*Vector-borne pathogens are emerging with increased frequency. This manuscript presents an interesting and potentially useful approach to the incrimination of vectors of novel emerging pathogens and I believe it to be worthy of publication. I have a couple of queries and suggestions, detailed below.*

*Results section, third paragraph: I suspect one criticism of this approach will be that it results in overly broad predictions (although even if true that would not mean it will not be useful in suggesting targets for epidemiological study). I would like the authors to add the result of applying the same method to 'predict' the vectors of dengue virus, yellow fever and West Nile virus when information on the respective viruses is removed from the training data.*

Following this suggestion and others, we conducted additional analyses to validate this method of prediction. We fit models to datasets omitting West Nile virus, yellow fever virus, and dengue virus individually to compare the models’ predictions against the “known outcome” of vectors of these viruses, as detailed above (subsections “Model Validation”).

Discussion section, first paragraph: could the authors elaborate on the additional regions or populations at risk from transmission based on their expanded 'worst case' vector list – or, even better, which newly-incriminated vectors make most difference to this? This should be possible using the dataset they have already collected (with the caveat that this is a worst-case scenario, and vector density and local environmental conditions also affect the potential for transmission).

We agree that prioritizing geographic regions or populations would be a helpful exercise for public health institutions, but we believe these conclusions are best made by public health experts who are able to incorporate vector biology, local climatic conditions, human exposure patterns, demographic variables, etc. We provide maps of species’ ranges and describe the additional geographic regions which may be at risk, should these candidate species be confirmed as vectors and supply sufficient vectorial capacity. By overlaying species ranges, we hope to demonstrate where transmission may be possible because of the presence of a competent vector, but not presently identified “high risk” areas. As you mention, there are many biotic, environmental and socio-economic local contexts that inform transmission dynamics, and we do not believe this to be a conclusion we can draw from our results. See Discussion paragraph four for further discussion of the limitations of our results.